# Clinicopathological Appearance of Epidermal Growth-Factor-Containing Fibulin-like Extracellular Matrix Protein 1 Deposition in the Lower Gastrointestinal Tract: An Autopsy-Based Study

**DOI:** 10.3390/ijms25147581

**Published:** 2024-07-10

**Authors:** Shojiro Ichimata, Yukiko Hata, Koji Yoshida, Naoki Nishida

**Affiliations:** Department of Legal Medicine, Faculty of Medicine, University of Toyama, Toyama 930-0194, Japannishida@med.u-toyama.ac.jp (N.N.)

**Keywords:** age-related amyloid, Congo red staining, constipation, epidermal growth-factor-containing fibulin-like extracellular matrix protein 1, lower gastrointestinal tract

## Abstract

This study examined the patterns of epidermal growth-factor-containing fibulin-like extracellular matrix protein 1 (EFEMP1) deposition in the small intestine and colon to evaluate the association between the histopathological severity of EFEMP1 deposition and constipation and determine the colocalization of amyloid transthyretin (ATTR) and EFEMP1 deposits. In 40 older cases (≥80 years of age), EFEMP1 deposition in the small intestine initiated in the submucosal and subserous vessels, subserous interstitium, and serosa (early stage), progressing to the muscularis propria and peri-Auerbach plexus area (intermediate stage), and finally spreading diffusely to other areas, excluding the mucosa and muscularis mucosa (advanced stage). The colon had a similar pattern of progression. During the middle-to-advanced stages, amyloid formation was observed in some vascular and serous deposits. A subgroup of cases was identified in which EFEMP1 deposition was the only presumed cause of constipation. Additionally, we demonstrated the colocalization of ATTR and EFEMP1 deposition. Apple-green birefringence was detected under polarized light only in approximately one-half of the cases in the small intestine and one-third of the cases in the colon. These findings strongly suggest that EFEMP1 deposits are correlated with pathological conditions of the lower gastrointestinal tract. As the histopathological diagnosis using Congo red-stained specimens is challenging, the combined use of elastic fiber staining and EFEMP1 immunohistochemistry is recommended to identify EFEMP1 deposition.

## 1. Introduction

Amyloid epidermal growth-factor-containing fibulin-like extracellular matrix protein 1 (AEFEMP1) amyloidosis is an age-related disorder [1,2]. Although it affects elastic fibers throughout the body, the most prevalent deposition is observed in the lower gastrointestinal tract [3]. We recently reported a case of granulomatous-type enterocolic lymphocytic phlebitis resembling amyloid-beta-related angiitis associated with EFEMP1/AEFEMP deposition and demonstrated its indirect pathogenicity [4,5]. However, in contrast to amyloid-beta [6,7], the direct pathological effects associated with EFEMP1/AEFEMP1 deposition are unclear [8].

Because amyloid deposits exhibit different deposition patterns depending on the precursor protein, understanding their morphology is essential for amyloid typing [9,10]. Although 42 amyloid precursor proteins are known [2], the pattern of deposition around elastic fibers is characteristic of AEFEMP1 [3]. However, the details of characteristic deposition patterns, especially which elastic fibers in which areas are most frequently affected, are unknown. Demonstrating Congo red-positive structures exhibiting apple-green birefringence under polarized light is crucial for the histopathological diagnosis of amyloid. However, it is unclear how often this can be demonstrated in EFEMP1 deposits, which show only weak congophilia [1].

Den Braber-Ymker et al. reported that intestinal involvement in amyloidosis, including amyloid-A-derived and immunoglobulin-light-chain-derived amyloidosis, is sequential, and involvement of the muscular layer and the subsequent loss of myenteric interstitial cells of Cajal may lead to dysmotility [11]. Because deposition in the muscular layer and around the Auerbach plexus is one of the characteristic deposition patterns of EFEMP1/AEFEMP1 [3], EFEMP1/AEFEMP1 deposition may cause dysmotility and constipation. In particular, AEFEMP1 deposition is closely associated with aging and is hypothesized to be a cause of lower gastrointestinal tract disorders, especially in older adults [1,2]. However, studies on the association between constipation and EFEMP1/AEFEMP1 deposition are lacking.

Amyloid formation and localization in a specific tissue may be triggered by fibril nuclei or seeds, a phenomenon known as seeding [12]. We have reported the co-deposition of amyloid transthyretin (ATTR) and amyloid atrial natriuretic factor—both age-related amyloidosis [13]—in the atrium [14], suggesting the presence of seeding between these amyloid deposits. Because ATTR deposition frequently involves the gastrointestinal tract [15], AEFEMP1 deposition may colocalize with ATTR deposition. According to Tasaki et al., ATTR and AEFEMP1 deposits do not colocalize in the colon [1]. However, as only one case was evaluated, the colocalization of the two amyloid deposits remains unclear.

Therefore, we conducted histopathological examinations of several samples from elderly cases (≥80 years old). We focused on the morphologies and distribution patterns of EFEMP1/AEFEMP1 deposits in each histological structure of the lower gastrointestinal tract. We examined the prevalence of cases where EFEMP1/AEFEMP1 deposition was presumed to be the cause of constipation. Moreover, we assessed whether colocalization of EFEMP1/AEFEMP1 and ATTR deposition is observed in the lower gastrointestinal tract.

## 2. Results

### 2.1. Clinical Profiles and Demographics

In total, 41 cases were identified. One case was excluded due to death by burning as the tissue was determined to be thermally denatured and unsuitable for analysis. Consequently, 40 cases remained eligible for histopathological study. Specimens of the colon and small intestine were available for 40 and 22 cases, respectively. Clinical and pathological data for these cases are summarized in Table 1. Appendix A shows detailed clinical and histopathological findings.

Overall, 5 patients in the small intestine case group and 10 in the colon case group had a medical history of constipation and/or were taking laxatives, including magnesium oxide, sennoside, and lubiprostone, and/or showed pseudomelanosis coli, which suggested the presence of constipation [16]. One patient had diverticulosis. One patient had incidental adenocarcinoma in the ascending colon, and one patient died due to perforation of the rectum due to adenocarcinoma. No statistically significant differences in clinical or histopathological data or cause of death were found between the two groups.

### 2.2. Pathological Findings of EFEMP1 and AEFEMP1

The results of semiquantitative analysis in the small intestine and colon are summarized in Table 2, and all findings are provided in Appendix A.

EFEMP1/AEFEMP1 deposition was observed around elastic fibers in blood vessels and the interstitium in both organs; vascular deposition was more severe in the subserosa than in the submucosa. EFEMP1 deposition in submucosal vessels, subserosal interstitium, and serosa was significantly greater in the small intestine than in the colon. The total immunohistochemistry (IHC) score was higher in the small intestine than in the colon, although differences were statistically insignificant. However, EFEMP1 deposition in the mucosal interstitium and around the Auerbach plexus was significantly greater in the colon than in the small intestine. Furthermore, the characteristic elastofibrotic lesion extending from the longitudinal muscular layer to the subserosa was observed more frequently in the colon than in the small intestine (Figure 1a–h).

Notably, apple-green birefringence was identifiable in the vascular and serosal deposits of the small intestine, while it was only identifiable in the vascular deposits in the colon.

### 2.3. Evaluation of the Relationship between ATTR and EFEMP1/AEFEMP1 Deposition

Representative microphotographs of ATTR and EFEMP1/AEFEMP1 deposits are shown in Figure 2.

In patients with cardiac ATTR amyloidosis (ATTR-CA), the deposition rate was 75% (positive in 3/4 cases) in the small intestine and 57% (positive in 4/7 cases) in the colon. In the small intestine and colon, ATTR deposition was identified mainly in the vessels (Figure 2a,b), and vascular involvement was more severe in the submucosa than in the subserosa. Most ATTR deposits were observed on small-sized arteries and exhibited stronger congophilia and clear apple-green birefringence than EFEMP1/AEFEMP1 deposits (Figure 2c,d). After screening with Congo red staining, double IHC for transthyretin and EFEMP1 was performed in two cases (cases 6 and 23), which revealed moderate or high levels of ATTR and EFEMP1/AEFEMP1 deposition. Although most ATTR and EFEMP1/AEFEMP1 deposits were observed separately in the double-IHC specimens (Figure 2e,f), colocalization of ATTR and EFEMP1/AEFEMP1 occurred in some veins in the small intestine and colon (Figure 2g,h).

### 2.4. Clinicopathological Features of Patients with Constipation

A comprehensive review of patients with confirmed or suspected constipation was conducted to assess contribution of EFEMP1/AEFEMP1 deposition to constipation. Table 3 presents a summary of the clinical information and pathological findings for cases of constipation.

Three cases received a clinical diagnosis of constipation (definite), whereas seven used laxatives and/or exhibited pseudomelanosis coli (possible). One case each had diabetes and chronic thyroiditis. ATTR deposition was not observed in the colon of any patient. Three cases presented with Lewy body disease (LBD), which was consistent with Braak’s LBD stage 4 or higher [17]. LBD pathology outside the brain was identified in all cases, and one case showed LBD pathology in the colon (Case 11). Representative findings of LBD pathology in the gastrointestinal tract are shown in Figure 3, and the distribution of LBD pathology outside the brain is summarized in Table 4.

We hypothesize that EFEMP1/AEFEMP1 deposition could cause constipation if the patient had a median or higher total IHC score of ≥8 in the colon. Considering all available information, EFEMP1/AEFEMP1 deposition was presumed to be the sole cause of constipation in 4/10 cases, and in all three patients with colonic elastofibrosis. However, the cause of constipation was indefinite in 2/10 cases.

## 3. Discussion

Based on the results of the histopathological examination, EFEMP1 deposition in the small intestine was observed to initiate in association with elastic fiber formation in the submucosal and subserosal vessels, subserosal interstitium, and serosa (early stage), progressing into the muscularis propria and peri-Auerbach plexus area (intermediate stage), and diffusely spreading to other areas, excluding the mucosa and muscularis mucosae (advanced stage). A similar progression pattern was noted in the colon, with deposition in the subserosa interstitium being considerably less than that in the small intestine. During the middle-to-advanced stages, deposits exhibiting characteristic degeneration of elastic fibers and amyloid formation were presumed to occur. Notably, despite assessing all layers with autopsy material, apple-green birefringence was detectable under polarized light in approximately half and one-third of the cases in the small intestine and colon, respectively. Thus, the histopathological diagnosis of AEFEMP1 amyloidosis presents significant challenges, especially in biopsy specimens where the sampling area is usually limited to the submucosal layer. The difficulty might be higher in the presence of inflammation [4]. To be aware of this deposition, it is crucial to identify the characteristic changes in the elastic plate through elastic fiber staining and confirm EFEMP1 with IHC. If the deposits exhibit weak congophilia and lack apple-green birefringence under polarized light, they should be designated “EFEMP1 deposition” and not “AEFEMP1 deposition”.

The direct pathogenicity of EFEMP1/AEFEMP1 remains unknown. Tasaki et al. reported cases of gastrointestinal bleeding possibly caused by AEFEMP1 deposition [1,8] that suggested an association between intestinal ischemia and vascular vulnerability. In a detailed histopathological study of a case with severe EFEMP1/AEFEMP1 deposits, we have reported that the deposits were found in all organs of the body and prominently in the lower gastrointestinal tract [3]. EFEMP1 colocalized with fine elastic fibers but not with large elastic structures such as the elastic lamina in the aorta [18]. *Efemp1^−^^/^^−^* mice exhibited reduced numbers of elastic fibers in the fascia [19]. Based on these findings, we hypothesize that EFEMP1 is most abundant around elastic fibers in the lower gastrointestinal tract; therefore, EFEMP1/AEFEMP1 deposition initiates around elastic fibers in the lower gastrointestinal tract and is most strongly impaired there. In this study, we suggest that EFEMP1/AEFEMP1 deposition is a potential cause of constipation. However, the link between histopathological severity and constipation remains unclear. Based on the results, it can be hypothesized that EFEMP1/AEFEMP1 deposition contributes to lower gastrointestinal dysfunction but may not sufficiently disable on its own to cause constipation. Figure 4 illustrates the dysfunctions and symptoms presumed to be caused by EFEMP1/AEFEMP1 based on our findings and other reports.

To understand the direct pathogenicity of EFEMP1/AEFEMP1 deposition, further analysis using autopsy or surgical specimens with detailed clinical information is essential.

Remarkably, EFEMP1 deposition accompanied by elastofibrosis was identified in the mucosa and outer layer of the muscularis propria to subserosa. This pathology was not readily apparent in hematoxylin and eosin (H&E)-stained specimens and was negative for Congo red staining. Consequently, the use of elastic fiber staining and EFEMP1 immunostaining is deemed indispensable for accurate diagnosis. Elastofibrosis in the gastrointestinal tract has been predominantly documented within polypoid lesions, although instances of diffuse nonpolypoid lesions have been reported [20,21]. The histopathology outlined in the report of Schiffman et al. aligns with the elastic fiber alterations associated with EFEMP1 deposition [22]. Thus, cases categorized as elastofibrosis might be instances where EFEMP1 deposition is the underlying cause. The pathological implications of elastofibrosis and the temporal relationship between elastic fiber proliferation and EFEMP1 deposition (i.e., which initiates the other) are unclear and necessitate further investigation.

To our knowledge, this report is the first report to show colocalization of ATTR and EFEMP1/AEFEMP1 deposits. However, instances of colocalization were limited in number, and the presence of synergistic interactions between these deposits was unclear. There is growing consensus that in the central nervous system, a combination of one or more proteinopathies (mixed pathology) frequently manifest in individuals with neurodegenerative diseases, demonstrating synergistic interactions between these deposits [7,23,24]. Because many proteinopathies stem from age-related amyloid deposition [13], assessing the mixed pathology of age-related proteins in organs beyond the brain may become increasingly vital. Considering that different amyloid precursor proteins often lead to distinct deposition patterns in various organs and forms of deposits, a comprehensive understanding of the deposition patterns of each amyloid type is imperative for accurate diagnosis.

In addition to certain bias in our study population, this study was constrained by incomplete clinical information for some patients, such as the presence or absence of constipation and medication history, primarily due to the absence of severe clinical symptoms. Particularly, there was a lack of detail and availability of diagnoses for neurodegenerative diseases, including LBD. However, we believe that our detailed neuropathological evaluation [25,26], which followed current diagnostic guidelines, had minimized the possibility that we might overlook any neurodegenerative diseases that could contribute to constipation. Detailed information was unavailable for the vaccination status and history of COVID-19 infection in the analyzed cases. Degeneration was more pronounced in muscularis mucosae in autopsy material than in surgical material, potentially impacting the immunoreactivity of EFEMP1 in this region. Compared to amyloid deposits derived from other precursor proteins, EFEMP1/AEFEMP1 deposits exhibited weak congophilia. The shape of the deposits varied across sites, necessitating the use of a complex semiquantitative grading system in this study. There was a lack of information on the histopathological deposition pattern of EFEMP1/AEFEMP1 deposits, and studies are warranted in the future. Consequently, it should be noted that the results of this study are only preliminary and do not directly prove the hypotheses presented in Figure 4. Further analysis using surgical specimens with lesser degeneration and more-complete clinical information is warranted. The small sample size, with only 40 cases, was another limitation of this study. The validity of this study would be better demonstrated with a larger sample size and a more diverse population.

In conclusion, this study presents a histopathological evolutionary pattern of EFEMP1/AEFEMP1 in the lower gastrointestinal tract, which is potentially associated with constipation in elderly adults. Furthermore, the findings revealed that EFEMP1/AEFEMP1 deposition colocalizes with ATTR deposition, although the colocalization of the two is presumed to be coincidental. Given the challenges of histopathological diagnosis on Congo red-stained specimens, we recommend the combined use of elastic fiber staining and IHC for EFEMP1 to prevent the overlooking of this deposition. Further analysis using cases with detailed clinical information is essential to understand the pathogenicity of EFEMP1/AEFEMP1 deposition and its relationship with other age-related amyloid deposits.

## 4. Materials and Methods

### 4.1. Case Selection

We reviewed the archives of all medical autopsy patients in our department from February 2020 to January 2022. First, we selected 164 cases in whom all organs, including the brain, could be sampled. Of these, we selected those ≥80 years of age, considering that EFEMP1/AEFEMP1 is an age-related condition [1,13]. We extracted cases where standard histopathologic studies based on H&E and elastica–Masson staining in general organs [27,28], neuropathologic studies based on Luxol fast blue/H&E staining and IHC [25], and Congo red staining and IHC-based amyloid typing were conducted in the heart [14]. Patients’ demographic and clinical characteristics (including cause of death) were retrieved from the medical records of police examinations and contributions from family members or from the primary physician if a record indicated clinic visits. This study was approved by the Ethical Committee of Toyama University (I2020006) and performed according to the ethical standards outlined in the 1964 Declaration of Helsinki and its 2008 amendment.

### 4.2. Tissue Samples

One block per organ was sampled from the lower gastrointestinal tract. Specimens were fixed in 20% buffered formalin and routinely embedded in paraffin. Then, 4 μm thick sections were cut and stained with H&E, elastica–Masson, or underwent IHC. Furthermore, 6 μm thick sections were cut and stained with phenol Congo red [29].

### 4.3. Semiquantitative Grading System for EFEMP1/AEFEMP1 Deposition

Representative microphotographs displaying the deposition patterns of EFEMP1/AEFEMP1 are presented in Figure 5.

The severity of immunohistochemical findings related to EFEMP1/AEFEMP1 deposition was assessed semiquantitatively. The severity of EFEMP1/AEFEMP1 pathology within vessels in the submucosa and subserosa and interstitium in each histological layer of the lower gastrointestinal tract, including the mucosa and muscularis mucosae, muscularis propria, thee area around the Auerbach plexus, subserosa (including mesentery), and serosa, was graded using a four-point scoring system as follows:**Vessel Grading:**Grade 0: No vascular EFEMP1 deposition.Grade 1: Occasional vessels with EFEMP1 deposition without amyloid properties, usually not occupying the thickness of the entire wall.Grade 2: A moderate number of vessels with EFEMP1 deposition, some occupying the full thickness of the wall and may exhibit focal amyloid properties.Grade 3: Many vessels with EFEMP1 deposition, most occupying the full thickness of the wall and exhibiting focal amyloid properties.
**Interstitium Grading:**Grade 0: No interstitial EFEMP1 deposition.Grade 1: A few EFEMP1 deposits in the interstitium occupying each low-power (×10 microscope objective) field.Grade 2: Moderate EFEMP1 deposits in the interstitium occupying each low-power (×10 microscope objective) field.Grade 3: Many EFEMP1 deposits in the interstitium occupy each low-power (×10 microscope objective) field, some exhibiting a massive and nodular deposition pattern.

Subsequently, the sum of all IHC deposition grades was calculated (total IHC score). Congo red-positive structures, demonstrating typical apple-green birefringence under polarized light, were histologically confirmed as amyloid deposits. The representative microphotographs of this semiquantitative grading system are presented in Figure 6.

The severity of ATTR deposition on the vessels was assessed using the same grading system.

### 4.4. Single and Double IHC

Single IHC was performed using primary antibodies against fibulin-3 (EFEMP1) (mouse, clone mab3-5, 1:2000, Santa Cruz, TX, USA), and phosphorylated α-synuclein (clone LB508, 1:500; Zymed, San Francisco, CA, USA) was performed in all cases. IHC was performed for prealbumin (transthyretin) (rabbit, clone EPR3219, 1:2000, Abcam, Cambridge, UK) in cases with positive ATTR-CA and a deposition pattern suspicious of ATTR deposition in the intestine and/or colon on Congo red-stained specimens. Antigen retrieval was performed using 98% formic acid for 1 min (EFEMP1 and transthyretin) or heat-mediated method using pH9 solution for 20 min (phosphorylated α-synuclein). Single IHC was performed using the Leica Bond-MAX automation system and Leica Refine detection kits (Leica Biosystems, Richmond, IL, USA), according to the manufacturer’s instructions. All sections were counterstained with hematoxylin.

In cases of suspected colocalization of ATTR and EFEMP1/AEFEMP1 deposition in the small intestine and/or colon, double IHC was performed using the antibodies listed. First, IHC for EFEMP1 was performed using the same procedure as that for single IHC. After the first IHC, sections were incubated with 0.3% H_2_O_2_ for 10 min and then incubated with primary antibodies against transthyretin (overnight, 4 °C). Signal was developed using the immunoenzyme polymer method (Histofine Simple Stain MAX PO Multi; Nichirei Biosciences, Tokyo, Japan) with the Vina Green Chromogen Kit (BioCore Medical Technologies, Gaithersburg, MD, USA) for 5 min. All sections were counterstained with hematoxylin.

### 4.5. Statistical Analysis

Data were analyzed using IBM SPSS statistics version 29 (SPSS Inc., Chicago, IL, USA), and the threshold for statistical significance was set at *p* < 0.05. Fisher’s exact test was used for categorical variables (presence of symptoms, pathological findings, and cause of death). Ordinal variables (pathological scores) were compared using the Mann–Whitney *U* test.

## Figures and Tables

**Figure 1 ijms-25-07581-f001:**
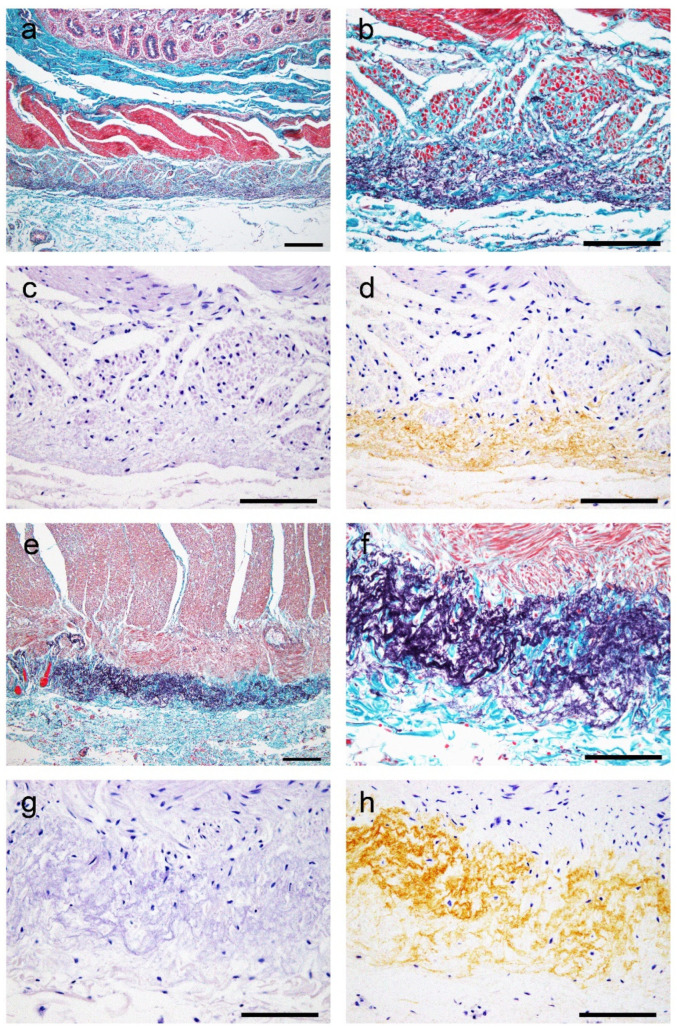
Representative microphotographs showing elastofibrosis of the muscularis propria and subserosa associated with EFEMP1 deposition. (**a**–**h**) Colon: (**a**,**b**,**e**,**f**) elastica–Masson staining; (**c**,**g**) Congo red staining; (**d**,**h**) IHC for EFEMP1. (**a**–**d**) 31 colon cases; (**e**–**h**) 37 colon cases. Panels (**b**–**d**) and panels (**f**–**h**) show the same sites in serial sections, respectively. (**a**–**d**) Elastofibrosis is observed in the outer part of the longitudinal muscular layer to the superficial part of the subserosa (**a**,**b**). In this lesion, congophilia was nearly negative, whereas obvious immunoreactivity for EFEMP1 was identified. (**e**–**h**) Proliferation of elastic fibers is obvious compared with the first case (**e**,**f**). However, congophilia and immunoreactivity for EFEMP1 are similar to the first case (**g**,**h**). Scale bar = 200 μm (**a**,**e**); 100 μm (**b**–**d**,**f**–**h**).

**Figure 2 ijms-25-07581-f002:**
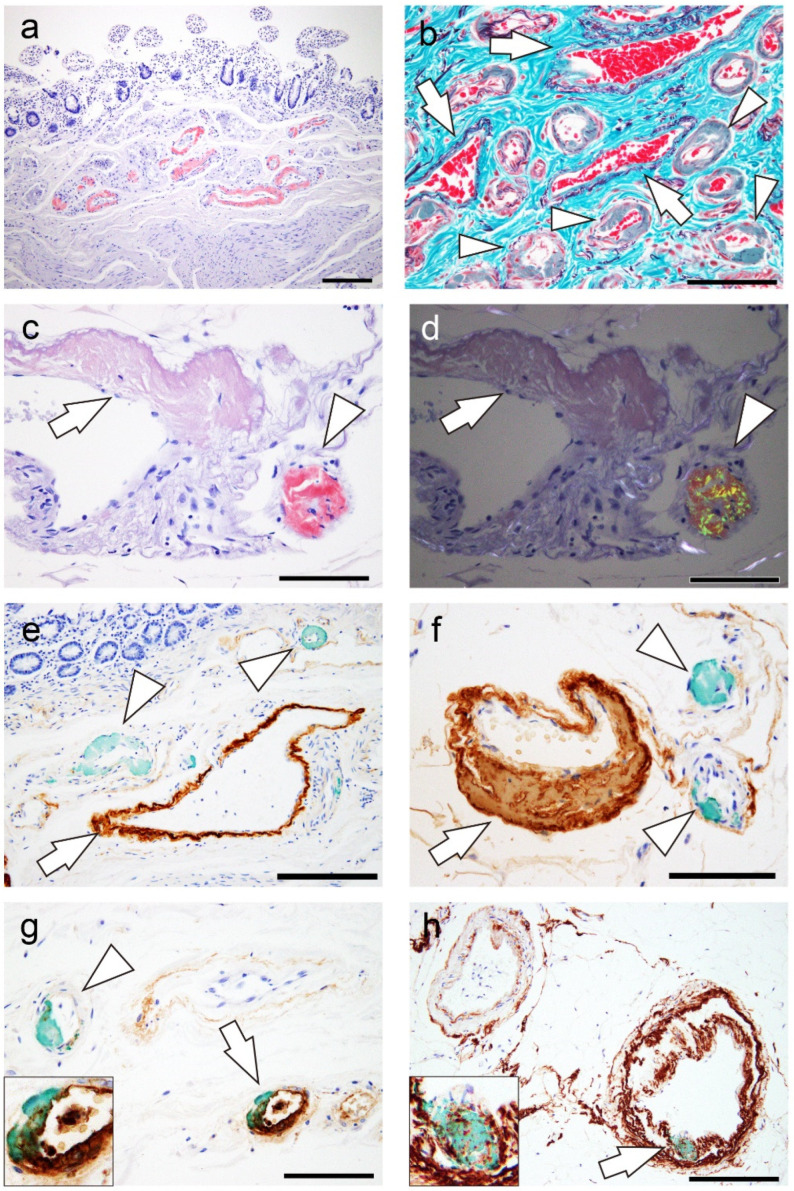
Representative microphotographs of ATTR and EFEMP1 deposition. (**a**–**f**) Small intestine; (**g**,**h**) colon. (**a**,**c**,**d**) Congo red staining under bright-field illumination (**a**,**c**) and under polarized light (**d**); (panel **a**, inset) IHC for transthyretin; (**e**–**h**) double IHC for transthyretin and EFEMP1. (**a**,**b**) ATTR deposition pattern in the submucosa. ATTR deposition is identified mainly in the vessels and involves relatively small arteries (arrowheads in panel **b**). Involvement of the veins (arrows in panel **b**) is not obvious. (**c**,**d**) ATTR deposition pattern in the subserosa. ATTR deposition (arrowhead) shows stronger congophilia than EFEMP1 deposition (arrow) and exhibits clear apple-green birefringence under polarized light (arrowhead in panel **d**). (**e**,**f**) Most ATTR and EFEMP1/AEFEMP1 deposits are observed independently (arrowheads and arrows indicate depositions of ATTR and EFEMP1/AEFEMP1, respectively). (**g**,**h**) However, colocalization of the two deposition types is identified in some vessels (arrowhead indicates ATTR deposition; arrow indicates ATTR and EFEMP1/AEFEMP1 deposition). Scale bar = 200 μm (**a**,**e**); 100 μm (**b**–**d**,**f**–**h**).

**Figure 3 ijms-25-07581-f003:**
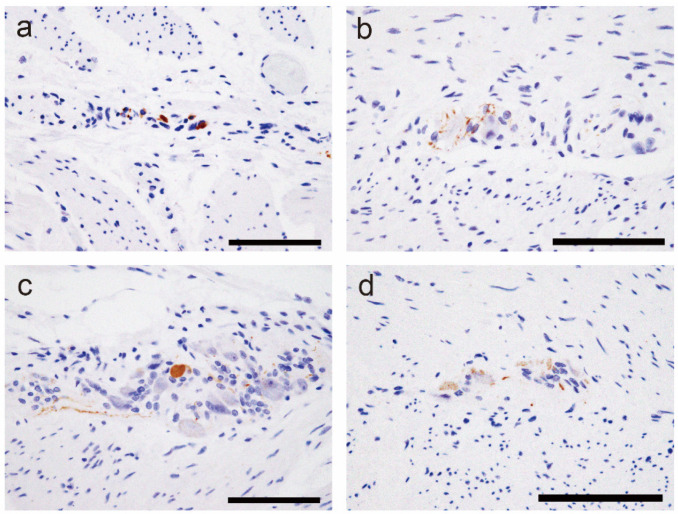
Representative microphotographs of alpha-synuclein deposition in the gastrointestinal tract. (**a**–**d**) Findings in Case 11: (**a**) Esophagus; (**b**) Stomach; (**c**) Small intestine; (**d**) Colon. Alpha-synuclein deposition is observed in the Auerbach plexus. Scale bar = 100 μm (**a**–**d**).

**Figure 4 ijms-25-07581-f004:**
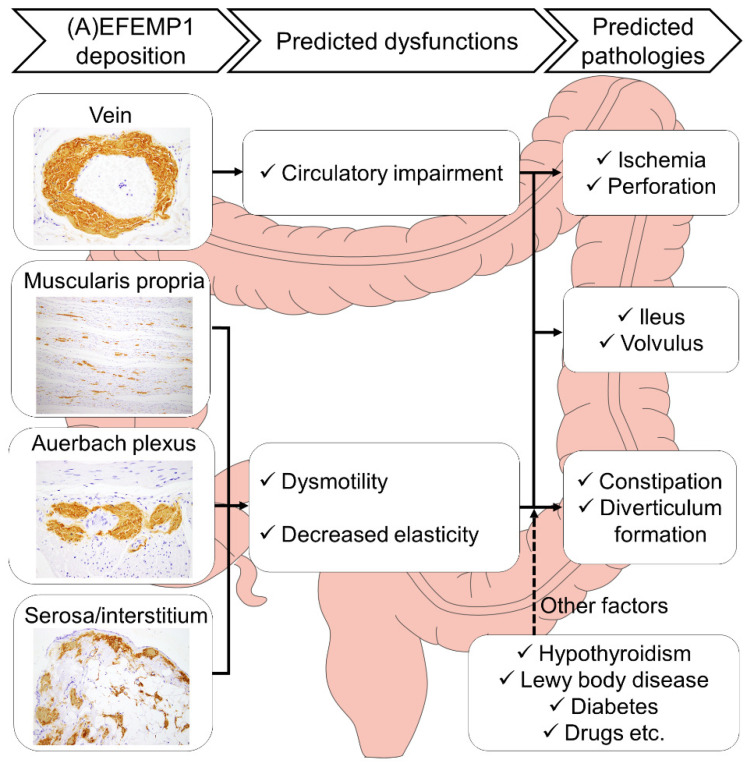
Schematic presentation of the hypothesized pathological effects of EFEMP1/AEFEMP1 deposition in the lower gastrointestinal tract. It is hypothesized that EFEMP1/AEFEMP1 deposition in blood vessels causes impaired blood flow, whereas deposition around the Auerbach plexus, muscularis propria, subserosa, and serosa reduces peristalsis and causes wall vulnerability. Complications such as hypothyroidism, LBD, and diabetes are presumed to be exacerbating factors in these conditions. A combination of these pathologies may result in ischemia of the intestinal tract and associated perforation, ileus, intestinal volvulus, constipation, and diverticulum formation.

**Figure 5 ijms-25-07581-f005:**
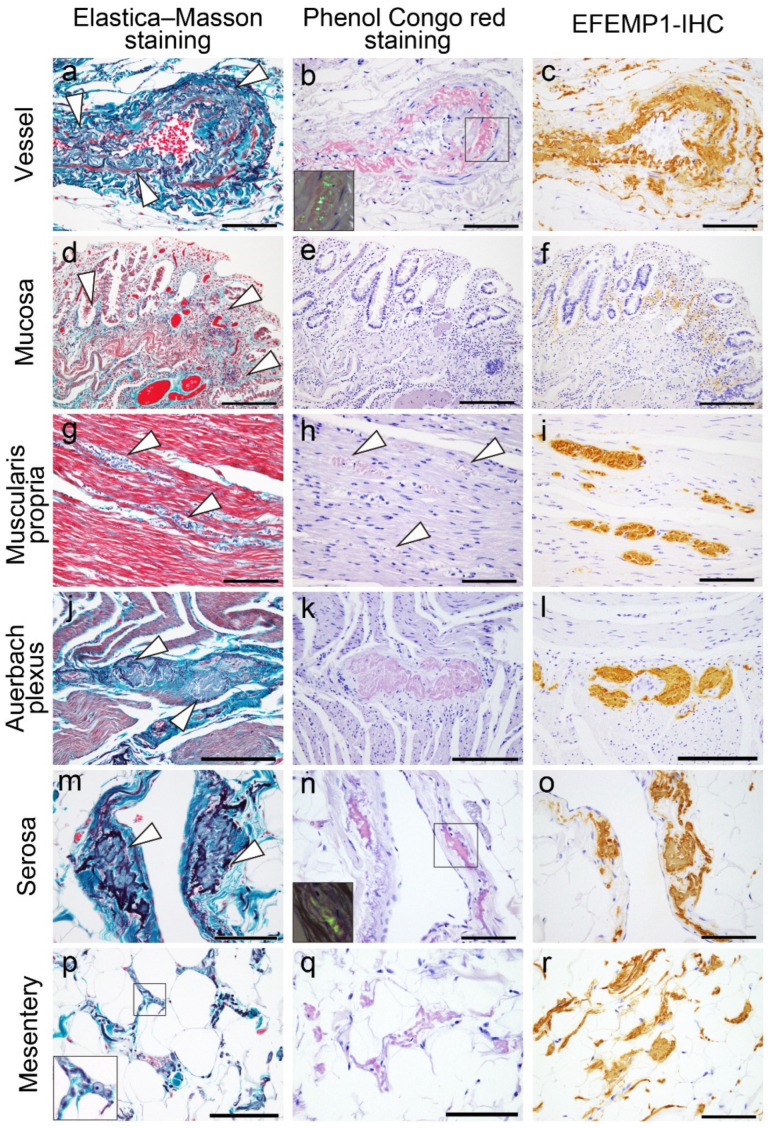
Representative microphotographs of EFEMP1 on elastica–Masson and Congo red staining and IHC for EFEMP1 in the lower gastrointestinal tract. Deposits in the vessel (**a**–**c**); mucosa and muscularis mucosae (**d**–**f**); muscularis propria (**g**–**i**); Auerbach plexus (**j**–**l**); serosa (**m**–**o**); and interstitium of the subserosa (including the mesentery) (**p**–**r**). (**a**,**d**,**g**,**j**,**m**,**p**) Elastica–Masson staining; (**b**,**e**,**h**,**k**,**n**,**q**) Congo red staining under bright-field illumination; (insets in (**b**,**n**)) Congo red staining under polarized light; (**c**,**f**,**i**,**l**,**o**,**r**) IHC for EFEMP1. Notably, areas positive for EFEMP1 deposition show proliferation (panel **d**), contortion, thickening, and fragmentation of elastic fibers (panels **a**,**g**,**j**,**m**,**p**). Deposits in the muscularis mucosae show no congophilia (**e**), and deposits in the muscularis propria (**h**), peri-Auerbach plexus (**k**), and interstitium of the subserosa (**q**) exhibit weak congophilia compared with that in the vessel (**b**) and serosa (**n**). Arrowheads indicate EFEMP1 deposits, and the area indicated by the black square is the region shown in the inset. Scale bar = 200 μm (**d**–**f**,**g**–**i**); 100 μm (**a**–**c**,**g**–**i**,**m**–**r**).

**Figure 6 ijms-25-07581-f006:**
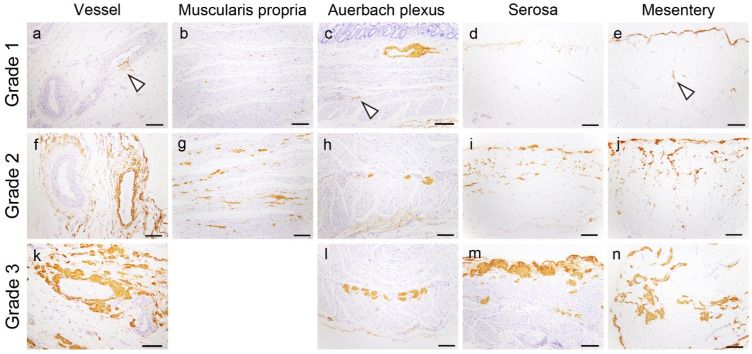
Representative microphotographs of EFEMP1 deposition grading used in this study: (**a**–**n**) IHC for EFEMP1; (**a**,**f**,**k**) vessel; (**b**,**g**) muscularis propria; (**c**,**h**,**l**) Auerbach plexus; (**d**,**i**,**m**) serosa; and (**e**,**j**,**n**) mesentery (interstitium). Arrowheads indicate EFEMP1 deposits. Scale bar = 200 μm (**a**–**n**).

**Table 1 ijms-25-07581-t001:** Clinical and pathological profiles of the participants.

	Small Intestine	Colon
# of cases	22	40
Sex (F/M)	11/11	20/20
Mean age (range), all	86.3 ± 5.0 (80–100)	85.9 ± 4.8 (80–100)
Mean age (range), female	86.5 ± 5.5 (80–100)	86.6 ± 5.3 (80–100)
Mean age (range), male	86.0 ± 4.5 (80–94)	85.2 ± 4.0 (80–94)
Constipation (%) *	5/18 (28)	10/34 (29)
Diabetes-positive (%) *	1/18 (6)	4/34 (12)
Hypothyroidism (%) **	2/18 (11)	2/34 (6)
Chronic thyroiditis (%)	2 (9)	3 (8)
Lewy body disease (%)	7 (32)	11 (28)
ATTR-CA-positive (%)	4 (18)	7 (18)
Cause of death (%)		
Illness	6 (27)	9 (23)
Accident	13 (59)	23 (58)
Suicide or homicide	3 (14)	8 (20)

Abbreviations: ATTR-CA, amyloid transthyretin cardiac amyloidosis; F, female; M, male. * Four and six cases were excluded from the small intestine and colon case groups, respectively, as the medical and medication history of these cases were not available. ** Patients with a clinical diagnosis of hypothyroidism and/or who were prescribed thyroid hormone medication (levothyroxine sodium hydrate).

**Table 2 ijms-25-07581-t002:** Summary of histopathological grading between the small intestine and colon.

	Small Intestine	Colon	*p* Value *
Mucosa-IHCG (range)	0.1 ± 0.3 (0–1)	0.5 ± 0.6 (0–2)	**0.01**
Submucosa-vessel-IHCG (range)	1.7 ± 0.7 (1–3)	1.3 ± 0.7 (0–3)	**0.02**
Muscularis propria-IHCG (range)	1.1 ± 0.8 (0–2)	1.3 ± 0.8 (0–3)	0.55
Auerbach plexus-IHCG (range)	0.8 ± 1.0 (0–3)	1.2 ± 0.9 (0–3)	**0.03**
Subserosa-vessel-IHCG (range)	2.1 ± 0.8 (1–3)	2.2 ± 0.7 (1–3)	0.96
Subserosa-interstitium-IHCG (range)	2.1 ± 0.7 (1–3)	1.2 ± 0.8 (0–3)	**<0.01**
Serosa-IHCG (range)	2.5 ± 0.7 (1–3)	0.9 ± 0.7 (0–3)	**<0.01**
Total IHC score (range/median)	10.4 ± 3.6 (4–17/10.5)	8.5 ± 3.7 (1–16/8.0)	0.09
Birefringence (%) **	12/22 (55)	14/40 (35)	0.18
Elastofibrosis (%)	1/22 (5)	7/40 (18)	0.24
ATTR-positive (%)	3 (14)	4 (10)	0.69

Abbreviations: ATTR, amyloid transthyretin; IHCG, immunohistochemical grading of epidermal growth-factor-containing fibulin-like extracellular matrix protein 1. **Boldface** signifies statistically significant values at *p* < 0.05. * Small intestine vs. colon, comparison using Fisher’s exact test or the Mann–Whitney U test. ** Number and percentage of cases with apple-green birefringence under polarized light.

**Table 3 ijms-25-07581-t003:** Summary of clinical and pathological data in patients with definite or possible constipation.

Case#	Age	Sex	Constipation	DM	HypoT	Thyroiditis	LBD^17^ *	T-IHC	EF	BF	Possible Cause
9	87	M	Definite	Pos	Neg	Neg	0	13	Pos	Pos	DM/(A)EFEMP1
11	85	F	Possible	Neg	Neg	Pos	5	2	Neg	Neg	LBD/HypoT
12	83	M	Possible	Neg	Neg	Neg	4	3	Neg	Neg	LBD
13	88	F	Possible	Neg	Neg	Neg	0	8	Neg	Pos	(A)EFEMP1
14	90	F	Possible	Neg	Neg	Neg	5	9	Neg	Pos	LBD/(A)EFEMP1
15	84	F	Possible	Neg	Neg	Neg	0	8	Neg	Neg	(A)EFEMP1
20	88	M	Definite	Neg	Neg	Neg	0	11	Pos	Pos	(A)EFEMP1
25	90	F	Definite	Neg	Neg	Neg	0	3	Neg	Neg	Indefinite
27	80	M	Possible	Neg	Neg	Neg	0	5	Neg	Neg	Indefinite
32	87	M	Possible	Neg	Neg	Neg	0	10	Pos	Neg	(A)EFEMP1

Abbreviations: BF, birefringence; DM, diabetes mellitus; EF, elastofibrosis; HypoT, hypothyroidism; Int, intermediate; LBD, Lewy body disease; Neg, negative; Pos, positive; T-IHC, total IHC score. * One case had a clinical diagnosis of LBD (Case 11), one case had a clinical diagnosis of Alzheimer’s disease (Case 14), and one case had no clinical diagnosis of neurodegenerative diseases (Case 12).

**Table 4 ijms-25-07581-t004:** Summary of the distribution of LBD pathology outside the brain.

Case#	Clinical Diagnosis	LBD Stage^17^	Heart *	SCG	SNT	AG	Esophagus	Stomach	Small Intestine	Colon
11	LBD	5	Pos	Pos	Pos	Pos	Pos	Pos	Positive	Pos
12	None	4	Pos	Pos	Pos	NA	Neg	Pos	NA	Neg
14	AD	5	Pos	Pos	Neg	Pos	Neg	Neg	NA	Neg

Abbreviations: AD, Alzheimer’s disease; AG, adrenal gland; SCG, superior cervical ganglion; SNT, sympathetic neural trunk; NA, not available. * The anterior ventricular septal region, superior vena cava region, and peri-sinoatrial node region were examined.

## Data Availability

The datasets used and analyzed in the current study are available from the corresponding authors upon request.

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
