# Peer review of "Clinicopathological Appearance of Epidermal Growth-Factor-Containing Fibulin-like Extracellular Matrix Protein 1 Deposition in the Lower Gastrointestinal Tract: An Autopsy-Based Study"

_ijms, 2024, doi:10.3390/ijms25147581_

Round 1
Reviewer 1 Report
Comments and Suggestions for Authors
The manuscript named “Clinicopathological appearance of epidermal growth factor containing fibulin-like extracellular matrix protein-1 (EFEMP1) deposition in the lower gastrointestinal tract: An autopsy-based study” by Ichimata et al. is dedicated to the association between histopathological severity of amyloid deposition in the small intestine and colon and constipation in old patients. The work also studied the colocalization of different amyloids, especially EFEMP1 and amyloid transthyretin (ATTR), the authors also studied the distribution of amyloid alpha-synuclein in the same zone in patients with Levi body disease. To colocalize these amyloids authors used immunostaining methods and amyloid-specific dyes. Analyzing their data, authors decided that EFEMP1 deposition was presumed to be the sole cause of constipation in 4/10 cases, and deposition was assumed to be the cause of constipation in all three patients with colonic elastofibrosis.
This reviewer liked the author's work very much. It is well-designed, well-done, and well-written in good English. Furthermore, it is an honest work, and the authors sincerely discussed all their data and the sustainability of their hypothesis. Figures are well presented, and the conclusions are clear and well supported. This article can be interesting to many researchers in the field.
This reviewer asks only minor changes to be made:
Related to text and Figures:
1. Line 150 Congo Red instead of Cong Red
2. Redistribution of text: In this reviewer's copy of the manuscript there are many blank spaces in the text because the figures and tables are improperly distributed. Can you, please, make the text more condensed ( it may be the question not to authors, but to editors).
Related to the colocalization:
1. While your work is specifically good it gives no general view. Please, add more general suggestions to the Discussion. Usually, if there is a presence of systemic amyloidosis, particular amyloid also accumulates in the intestines and colon (Alshehri, Hussein, 2020). In Alzheimer’s patients, beta-amyloid as well accumulates in the intestines near blood vessel walls (Jin et al, 2023). The accumulation of amyloids usually happens in the tissues with restricted zones, under the barriers like the brain-blood barrier, skin-vascular, or gut-vascular barrier (Ono et al, 2017). These barriers define paracellular permeability, as not allowing phagocytes to enter the tissue and producing a “no-cleanup” zone, shifting the balance between accumulation and removal of extracellular debris, including amyloid depositions (Inyushin et al, 2020).
The question arises, why EFEMP1 is accumulating in the small intestine and colon? Is it the manifestation of some systemic disease? As I understand, ATTR is accumulating only in patients with systemic ATTR amyloidosis. Can you, please, discuss why you studied specifically EFEMP1, and why it is involved in many patients? Please, add your suggestions to the discussion, it may help many researchers better understand the problem.
After these minor changes, this reviewer thinks the article may be accepted.
Alshehri SA, Hussein MRA. Primary Localized Amyloidosis of the Intestine: A Pathologist Viewpoint. Gastroenterology Res. 2020 Aug;13(4):129-137. doi: 10.14740/gr1303.
Jin J, Xu Z, Zhang L, Zhang C, Zhao X, Mao Y, Zhang H, Liang X, Wu J, Yang Y, Zhang J. Gut-derived β-amyloid: Likely a centerpiece of the gut-brain axis contributing to Alzheimer's pathogenesis. Gut Microbes. 2023 Jan-Dec;15(1):2167172. doi: 10.1080/19490976.2023.2167172.
Inyushin M, Zayas-Santiago A, Rojas L, Kucheryavykh L. On the Role of Platelet-Generated Amyloid Beta Peptides in Certain Amyloidosis Health Complications. Front Immunol. 2020 Oct 2;11:571083. doi: 10.3389/fimmu.2020.571083.
Ono S, Egawa G, Kabashima K. Regulation of blood vascular permeability in the skin. Inflammation Regen (2017) 37:11. doi: 10.1186/s41232-017-0042-9
Author Response
Response to Reviewer #2
We thank the Reviewer for their kind review and comments, which have helped us substantially improve our manuscript. In the revised manuscript, additions are indicated in red font.
The manuscript named “Clinicopathological appearance of epidermal growth factor containing fibulin-like extracellular matrix protein-1 (EFEMP1) deposition in the lower gastrointestinal tract: An autopsy-based study” by Ichimata et al. is dedicated to the association between histopathological severity of amyloid deposition in the small intestine and colon and constipation in old patients. The work also studied the colocalization of different amyloids, especially EFEMP1 and amyloid transthyretin (ATTR), the authors also studied the distribution of amyloid alpha-synuclein in the same zone in patients with Levi body disease. To colocalize these amyloids authors used immunostaining methods and amyloid-specific dyes. Analyzing their data, authors decided that EFEMP1 deposition was presumed to be the sole cause of constipation in 4/10 cases, and deposition was assumed to be the cause of constipation in all three patients with colonic elastofibrosis.
This reviewer liked the author's work very much. It is well-designed, well-done, and well-written in good English. Furthermore, it is an honest work, and the authors sincerely discussed all their data and the sustainability of their hypothesis. Figures are well presented, and the conclusions are clear and well supported. This article can be interesting to many researchers in the field.
This reviewer asks only minor changes to be made:
Related to text and Figures:
- Line 150 Congo Red instead of Cong Red
- Redistribution of text: In this reviewer's copy of the manuscript there are many blank spaces in the text because the figures and tables are improperly distributed. Can you, please, make the text more condensed ( it may be the question not to authors, but to editors).
Related to the colocalization:
- While your work is specifically good it gives no general view. Please, add more general suggestions to the Discussion. Usually, if there is a presence of systemic amyloidosis, particular amyloid also accumulates in the intestines and colon (Alshehri, Hussein, 2020). In Alzheimer’s patients, beta-amyloid as well accumulates in the intestines near blood vessel walls (Jin et al, 2023). The accumulation of amyloids usually happens in the tissues with restricted zones, under the barriers like the brain-blood barrier, skin-vascular, or gut-vascular barrier (Ono et al, 2017). These barriers define paracellular permeability, as not allowing phagocytes to enter the tissue and producing a “no-cleanup” zone, shifting the balance between accumulation and removal of extracellular debris, including amyloid depositions (Inyushin et al, 2020).
The question arises, why EFEMP1 is accumulating in the small intestine and colon? Is it the manifestation of some systemic disease? As I understand, ATTR is accumulating only in patients with systemic ATTR amyloidosis. Can you, please, discuss why you studied specifically EFEMP1, and why it is involved in many patients? Please, add your suggestions to the discussion, it may help many researchers better understand the problem.
After these minor changes, this reviewer thinks the article may be accepted.
Response:
We thank the Reviewer for their insights.
- We have corrected the typo (line 163).
- The arrangement of the text, figures, and tables will be coordinated with the editorial office after final acceptance.
- We have included a discussion on the reason of our focus on EFEMP1/AEFEMP1 (lines 266–274). However, we could not determine why EFEMP1 was found in many patients. We hypothesize that similar to other age-related amyloid depositions, an increase in local protein concentration resulted from age-related metabolic deterioration, leading to deposition.
Reviewer 2 Report
Comments and Suggestions for Authors
1) The authors present a good and systematic overview of use of histological techniques in the context of detection systemic amyloidosis and related protein aggregation patterns. New molecular fluorescent amyloid tools have been developed recently, for example in studies of insulin amyloidosis of diabetes patients. (See e.g. https://pubs.rsc.org/en/content/articlelanding/2020/ra/d0ra07742a)
It would be interesting to see a similar study including such fluorescent amyloid markers and if this might increase the sensitivity, not only in the colon, but also in other organs, such as heart and liver..
2) There are discussions in the scientific community if spike proteins/mRNA c19 vaccines can seed and induce amyloidosis. (See e.g.: https://www.news-medical.net/news/20230905/SARS-CoV-2-spike-protein-could-be-speeding-up-Alzheimers-and-other-brain-diseases-says-new-study.aspx)
Is the vaccination status know of the patients. It was noted the samples were collected in the range Feb 2020 -> 2022 so perhaps not too many patients were inoculated or had a known covid history. I urge the authors to comment on this in the introduction and/in the details of the patient history, etc.
Author Response
Response to Reviewer #3
We thank the Reviewer for their kind review and comments, which have helped us substantially improve our manuscript. The additions in the revised manuscript are indicated in red font.
1) The authors present a good and systematic overview of use of histological techniques in the context of detection systemic amyloidosis and related protein aggregation patterns. New molecular fluorescent amyloid tools have been developed recently, for example in studies of insulin amyloidosis of diabetes patients. (See e.g. https://pubs.rsc.org/en/content/articlelanding/2020/ra/d0ra07742a)
It would be interesting to see a similar study including such fluorescent amyloid markers and if this might increase the sensitivity, not only in the colon, but also in other organs, such as heart and liver..
2) There are discussions in the scientific community if spike proteins/mRNA c19 vaccines can seed and induce amyloidosis. (See e.g.: https://www.news-medical.net/news/20230905/SARS-CoV-2-spike-protein-could-be-speeding-up-Alzheimers-and-other-brain-diseases-says-new-study.aspx)
Is the vaccination status know of the patients. It was noted the samples were collected in the range Feb 2020 -> 2022 so perhaps not too many patients were inoculated or had a known covid history. I urge the authors to comment on this in the introduction and/in the details of the patient history, etc
Response:
We thank the Reviewer for their suggestions.
- Although we have not used the suggested fluorescent dyes, we have reported that this deposit stains positive with thioflavin S staining [Pathol Int. 2024;74(3):146-153. https://doi.org/10.1111/pin.13405].
- Unfortunately, we do not have detailed information on the vaccination status or history of COVID-19 infection in the analyzed cases. We have included this information in the revised manuscript (lines 325–326).